# Lipid-associated PML structures assemble nuclear lipid droplets containing CCTα and Lipin1

Jonghwa Lee[1], Jayme Salsman[2], Jason Foster[1], Graham Dellaire[1,2], Neale D Ridgway[1,3]

**Nuclear lipid droplets (nLDs) form on the inner nuclear membrane by a mechanism involving promyelocytic leukemia (PML), the protein scaffold of PML nuclear bodies. We report that PML structures on nLDs in oleate-treated U2OS cells, referred to as lipid-associated PML structures (LAPS), differ from canonical PML nuclear bodies by the relative absence of SUMO1, SP100, and DAXX. These nLDs were also enriched in CTP:phosphocholine cytidylyltransferase α (CCTα), the phosphatidic acid phosphatase Lipin1, and DAG. Translocation of CCTα onto nLDs was mediated by its α-helical M-domain but was not correlated with its activator DAG. High-resolution imaging revealed that CCTα and LAPS occupied distinct polarized regions on nLDs. PML knockout U2OS (PML KO) cells lacking LAPS had a 40–50% reduction in nLDs with associated CCTα, and residual nLDs were almost devoid of Lipin1 and DAG. As a result, phosphatidylcholine and triacylglycerol synthesis was inhibited in PML KO cells. We conclude that in response to excess exogenous fatty acids, LAPS are required to assemble nLDs that are competent to recruit CCTα and Lipin1.**

## Introduction

Lipid droplets (LDs) are cellular organelles composed of a core of triacylglycerol (TAG) and cholesterol ester (CE) surrounded by a monolayer of phospholipids and associated proteins. In response to nutrient and hormonal signals, fatty acids and cholesterol are stored in or released from LDs to provide energy and biosynthetic precursors as well as buffer the cell from fatty acid toxicity (Sztalryd & Brasaemle, 2017). The largest neutral lipid storage depot is in adipocytes but hepatocytes, enterocytes, and macrophages also have the capacity for short-term storage and release of fatty acids from LDs. Accumulation of LDs in tissues, caused by an imbalance between lipid storage and hydrolysis, is linked to pathological conditions such as hepatosteatosis, obesity, and lipodystrophy (Walther & Farese, 2012).

LDs are proposed to form in the ER by a process that requires the coordinated synthesis of TAG and phospholipids (Henne et al, 2018).

TAG initially forms a "lens" in the ER bilayer, expands, and pinches off into the cytoplasm and is coated with ER-derived phospholipids, which regulate surface tension and the storage capacity of LDs (Krahmer et al, 2011). Phosphatidylcholine (PC) is the most significant component of the surface monolayer of LDs (Tauchi-Sato et al, 2002; Bartz et al, 2007), and de novo PC synthesis by the cytidine diphosphate (CDP)-choline pathway in the ER is required for LD biogenesis (Krahmer et al, 2011; Aitchison et al, 2015). In mammalian cells, the CDP-choline pathway is regulated by CTP:phosphocholine cytidylyltransferase (CCT) α and β isoforms that are activated by translocation to nuclear and cytoplasmic membranes, respectively, in response to the content of PC and specific lipid activators, such as fatty acids or the type II conical-shaped lipids DAG and phosphatidylethanolamine (PE) (Arnold & Cornell, 1996; Arnold et al, 1997; Xie et al, 2004). During LD biogenesis in fatty acid-treated cells (Lagace & Ridgway, 2005; Gehrig et al, 2009) and during adipocyte differentiation (Aitchison et al, 2015), PC synthesis was increased by CCTα translocation from the nucleoplasm to the inner nuclear membrane (INM). In oleate-treated insect cells, ectopically expressed CCTα and the insect homologue CCT1 translocated from the nucleus to the surface of cytoplasmic LDs (cLDs) resulting in increased PC synthesis to facilitate LD expansion and TAG storage (Guo et al, 2008; Krahmer et al, 2011; Payne et al, 2014). In contrast, mammalian CCTα can exit the nucleus under some conditions (Northwood et al, 1999; Gehrig et al, 2009) but does not localize to the surface of cLDs in adipocytes and other cultured cells (Aitchison et al, 2015; Haider et al, 2018).

In addition to the INM, CCTα translocation to nuclear LDs (nLDs) can activate PC synthesis in oleate-treated Huh7 hepatoma cells (Soltysik et al, 2019). nLDs account for ≈10% of the total cellular LD pool in liver sections and hepatocytes and have a unique lipid and protein composition (Uzbekov & Roingeard, 2013; Lagrutta et al, 2017). In hepatocytes, nLDs can arise from TAG-rich droplets in the ER lumen that are precursors for very low density lipoprotein (VLDL) assembly (Soltysik et al, 2019). Instead of being incorporated into secreted VLDL, ER luminal LDs migrate into invaginations of the INM, termed as the type I nucleoplasmic reticulum (NR), and are released as nascent nLDs into the nucleoplasm where they could increase in size by fusion with other nLDs or by de novo TAG

[1]Department of Biochemistry and Molecular Biology, Dalhousie University, Halifax, Canada   [2]Department of Pathology, Dalhousie University, Halifax, Canada   [3]Department of Pediatrics, Dalhousie University, Halifax, Canada

Correspondence: dellaire@dal.ca; nridgway@dal.ca

synthesis (Ohsaki et al, 2016; Soltysik et al, 2019). During or after release into the nucleoplasm, nLDs associate with promyelocytic leukemia (PML) protein, in what were at the time referred to as PML nuclear bodies (PML NBs) (Ohsaki et al, 2016). PML NBs are dynamic subnuclear domains that regulate gene expression and stress responses and are strongly associated with proteins modified by the small ubiquitin modifier 1 (SUMO1) (Dellaire & Bazett-Jones, 2004; Ching et al, 2005; Van Damme et al, 2010), the most prominent being the SP100 nuclear antigen and the death-associated domain protein 6 (DAXX) (Zuber et al, 1995; Ishov et al, 1999). However, of the seven PML isoforms, only PML-II was associated with nLDs and required for their formation (Ohsaki et al, 2016). Moreover, the presence of CCTα and perilipin-3 on nLDs suggests that these lipid-associated PML-containing domains can regulate lipid synthesis (Soltysik et al, 2019), possibly to generate a metabolic signal to accommodate the uptake and storage of excess fatty acids.

Here, we report that the PML structures on nLDs in oleate-treated U2OS cells are deficient of SUMO1, SP100, and DAXX, proteins that are commonly associated with PML NBs. Therefore, to distinguish these structures from canonical PML NBs and reflect their association with nLDs and the lipid biosynthetic enzymes CCTα and Lipin1, we have designated them as lipid-associated PML structures (LAPS). Disrupting LAPS by CRISPR/Cas9 knockout of the *PML* gene in human U2OS osteosarcoma cells (PML KO) resulted in fewer nLDs that were deficient in CCTα, Lipin1, and DAG, resulting in decreased PC and TAG synthesis. These findings support the concept that LAPS are required for assembly of nLDs and nuclear lipid synthesis and that nLDs could provide a regulatory platform to orchestrate the cellular response to excess fatty acid uptake and storage.

# Results

### nLDs associate with non-canonical LAPS

Similar to hepatoma cells, U2OS osteosarcoma cells contain abundant nLDs that are associated with mCherry-PML-II (Ohsaki et al, 2016). As further evidence of their nucleoplasmic localization, nLDs in oleate-treated U2OS cells have endogenous PML and CCTα on their surface (Fig S1), and PML-positive nLDs were not encapsulated by the nuclear envelope (NE) based on immunostaining for emerin (Fig S2). Thus, U2OS cells offer an alternate and complementary model to investigate the structure and function of nLDs. Initially, we investigated whether the PML structures on nLDs differ from canonical PML NB by immunostaining control and oleate-treated U2OS cells for the PML NB–associated proteins SUMO1, SP100, and DAXX (Figs 1, S3, and S4). U2OS cells contain numerous PML NBs that are all positive for SUMO1, consistent with the essential role of SUMOylation in PML NB assembly and function (Zhong et al, 2000) (Fig 1A). Oleate treatment caused the loss of SUMO1-positive PML NB puncta and the appearance of PML-positive nLDs with different levels of SUMO1 expression (Fig 1B). When quantified, a weak (<50% of PML signal intensity) or non-existent SUMO1 signal was detected in 75% of PML-positive nLDs (Fig 1C). In addition, DAXX and SP100, proteins whose interaction with canonical PML NBs is SUMO-dependent (Ishov et al, 1999) or

constitutive (Sternsdorf et al, 1997), respectively, were strongly localized to PML NBs in untreated cells but weakly associated or absent in 80% of PML-positive nLDs in oleate-treated U2OS cells (Figs 1C, S3, and S4). Since the PML structures on nLDs are part of a large lipid complex and relatively devoid of canonical PML NB proteins, we propose that they be designated as LAPS.

To investigate how LAPS influence the composition and biogenesis of nLDs, we used U2OS cells in which the *PML* gene was knocked out (PML KO) by CRISPR/Cas9 gene editing leading to loss of expression of all PML isoforms (Attwood et al, 2019). CCTα or PML expression in U2OS cells was unchanged by oleate treatment, and CCTα expression was unaffected by PML KO (Fig 2A). In oleate-treated U2OS cells, CCTα was expressed in the nucleoplasm and on the surface of nLDs (Fig 2B), whereas, in PML KO cells, there were fewer BODIPY- and CCTα-positive nLDs, and CCTα was partially localize to the NE. Compared with control cells, the number of cLDs in PML KO cells was reduced slightly (Fig 2C). However, PML KO cells had a significant reduction in the number of nLD per cell (Fig 2D) and the percentage of CCTα-positive nLDs (Fig 2E). The cross-sectional area of cLDs in PML KO cells was similar to controls (Fig 2F), but there was a significant shift in the distribution toward small nLDs in PML KO cells (Fig 2G) that was reflected in a 40% decrease in average area (Fig 2H). CCTα-positive nLDs in control and PML KO cells were similar in size, suggesting the enzyme preferentially associates with larger nLDs that are formed by a PML-independent mechanism.

To identify the PML isoform involved in nLD formation in U2OS cells, PML KO cells were individually transfected with GFP-tagged version of the seven PML isoforms. In agreement with results in Huh7 cells (Ohsaki et al, 2016), GFP-PML-II associated with nLDs and increased their abundance in PML KO cells, whereas other PML isoforms did not associate or correct the PML KO phenotype (Fig S5A and B, results not shown). GFP-PML-II expression in PML KO cells significantly increased the number and size of nLDs to the level observed in wild-type U2OS cells (Fig S5C and D) and also restored the slight reduction in these parameters for cLDs (Fig S5E and F). The specific requirement for PML-II in nLD formation is further evidence that LAPS are a unique nuclear PML subdomain.

### CCTα and LAPS localize to discrete regions of nLDs

CCTα associates with membranes via its amphipathic α-helical M-domain, which is antagonized by phosphorylation of 16 serine, threonine, and tyrosine residues in the adjacent P-domain (Fig 3A) (Cornell & Ridgway, 2015). The binding of CCTα to LDs involves large hydrophobic side chains in the M-domain that insert into voids in the phospholipid monolayer (Prevost et al, 2018). To determine if the M- and P-domains regulate binding to nLDs, CCTα with truncations and point mutations in these domains was expressed in oleate-treated U2OS cells (Fig 3A and B), and localization on nLDs was quantified by immunofluorescence microscopy (Fig 3C and D). CCTα localization to nLDs was completely prevented by mutation of eight lysine residues in the M-domain (CCTα-8KQ) that form electrostatic interactions with membrane lipids (Johnson et al, 2003). Conversely, CCTα-3EQ, an M-domain mutant with enhanced membrane association (Johnson et al, 2003), was localized to nLDs as well as the NE. Deletion of the P-domain (CCTα-ΔP) and a dephosphorylated mimic with 16 serine residues mutated to

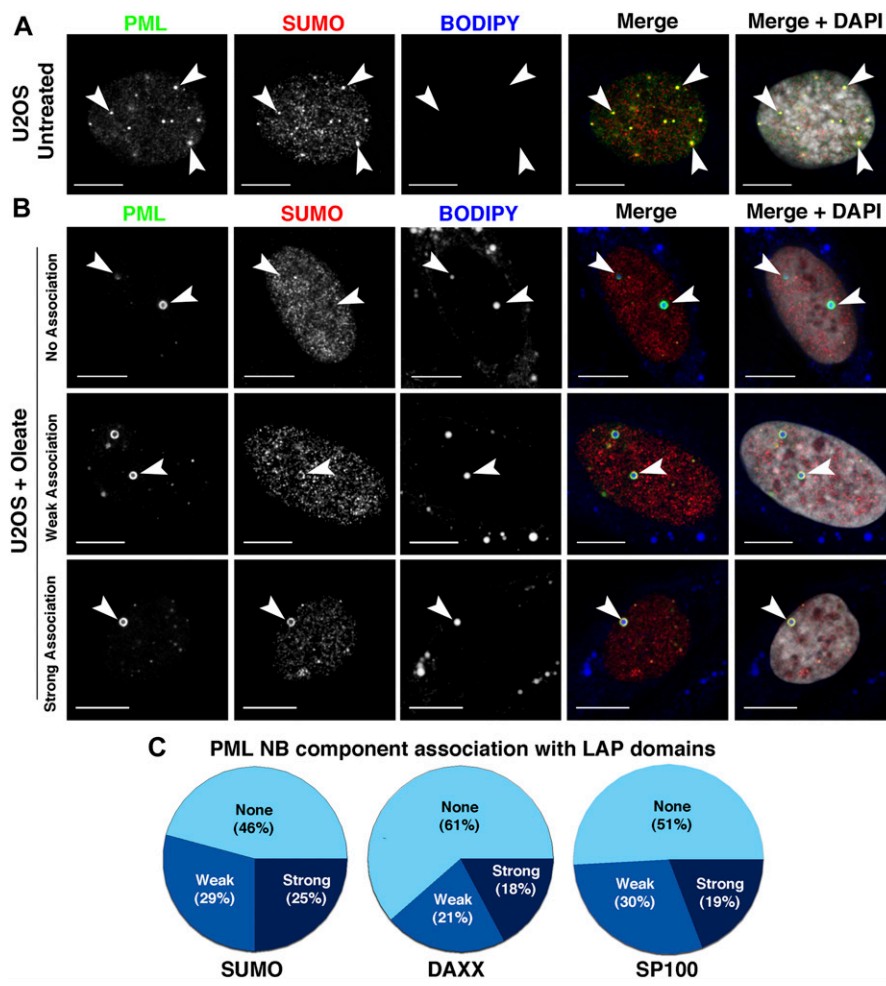

**Figure 1.  Lipid-associated PML structures (LAPS) on nuclear lipid droplets (nLDs) are deficient in PML nuclear body (NB) resident proteins.**
**(A, B)** U2OS cells were untreated (panel A) or incubated with oleate (400 μM) for 24 h (panel B) before immunostaining for PML (green) and SUMO1 (red) and imaging by confocal microscopy. LDs and nuclei were visualized with BODIPY 493/503 and DAPI, respectively (bar, 10 μm). **(B)** Examples of SUMO1 association (none, weak, or strong) with PML-positive nLDs are shown (panel B). **(C)** The intensity of SUMO1, DAXX, and SP100 association with PML-positive nLDs was scored as none, weak, or strong (described in the Materials and Methods section) for at least 100–150 nLDs from at least 50–100 cells. Examples of DAXX and SP100 localization with PML-positive nLDs are shown in Figs S3 and S4.

alanine (CCTα-16SA) were strongly associated with nLDs, whereas the phosphorylated mimic with 16 serine residues mutated to glutamate (CCTα-16SE) was not detected on nLDs. Catalytic dead CCTα-K122A was localized on nLDs similarly to the wild type. These results indicate that interaction of CCTα with the nLD phospholipid monolayer is mediated by electrostatic interactions with the M-domain and antagonized by phosphorylation of the P-domain.

Next, we used spinning disk confocal 3D and super-resolution radial fluctuation (SRRF) (Gustafsson et al, 2016) imaging to determine how endogenous CCTα and LAPS are organized on the surface of nLDs (Fig 4). 3D reconstruction of a typical nLD revealed a lipid body (BODIPY) located close to the basal surface of the INM (Fig 4A, a, b, and c). CCTα coated most of the nLD but was absent from the top of the particle (Fig 4A, d), which was occupied by a "cap" of PML protein (Fig 4A, e and f). SRRF imaging of this nLD in a cross section where CCTα and PML intersect (indicted by the yellow line in Fig 4A, b–f) revealed a punctate distribution (Fig 4B and C) consistent with each protein occupying non-overlapping, inter-digitated regions on the nLD surface (error and resolution analysis using NanoJ-SQUIRREL [Culley et al, 2018] for this image is shown in Fig S6). Additional 3D renderings of eight nLDs are shown in Fig S7. In the case of smaller nLDs (Fig S7A, object 1; Fig S7C, objects 5 and 6),

PML formed a patch on the surface that minimally overlapped with CCTα. Similar to Fig 4, large nLDs displayed a polarized PML patch (Fig S7A, object 2; Fig S7B, object 3). More infrequent were large nLDs with PML coating much of the surface and reduced CCTα association (Fig S7B, object 4). In summary, LAPS generally occupy only part of the surface of an nLD and have limited overlap with CCTα, which more strongly associates with large nLDs via its M-domain.

## LAPS regulate the DAG and Lipin1 content of nLDs

In addition to being a precursor for the TAG and PC that is incorporated into LDs, DAG also stimulates translocation and activation of CCTα on membranes (Arnold et al, 1997). Thus, we investigated whether CCTα interaction with nLDs could be regulated by DAG as a mechanism to coordinate PC and TAG synthesis. Initially, we visualized DAG in cells cultured with or without oleate for 24 h using the DAG biosensor GFP-C1(2)δ (Codazzi et al, 2001) (Fig 5). In untreated wild-type and U2OS PML KO cells, the DAG biosensor was localized on reticular and punctate structures in the cytoplasm (Fig 5A). Exposure of wild-type U2OS cells to oleate resulted in the appearance of the DAG biosensor on dispersed cytoplasmic

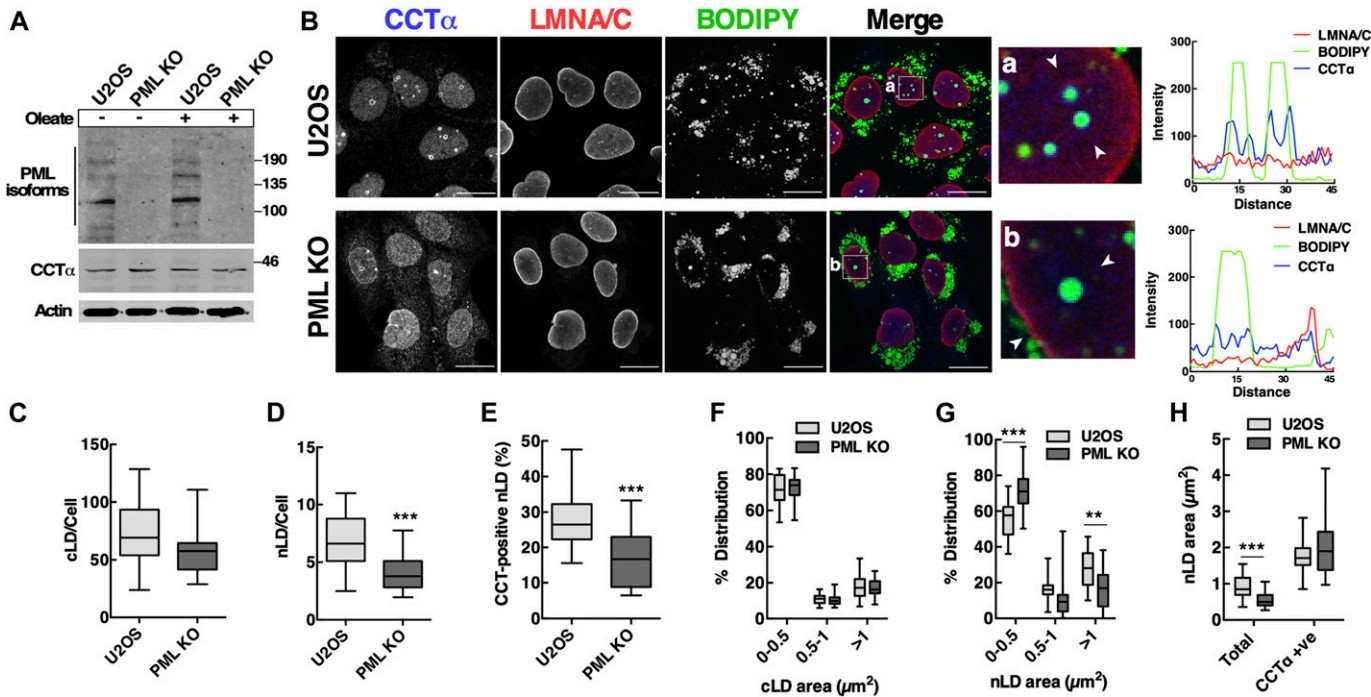

**Figure 2. Lipid-associated PML structures regulate biogenesis of nuclear lipid droplets (nLDs) and association of CCTα.**
**(A)** Lysates of U2OS and PML KO cells that were untreated or incubated with oleate (400 $\mu M$) for 24 h were immunoblotted with PML-, CCTα-, and actin-specific antibodies. **(B)** U2OS and PML KO cells were immunostained for CCTα and LMNA/C, and LDs were visualized with BODIPY 493/503 (bar, 20 $\mu m$). Arrows in magnified panels (a) and (b) indicate the regions for RGB line scan plots showing the localization of CCTα on the surface of nLDs. **(C, D)** Quantitation of cytoplasmic (cLDs) and nLDs in cells treated with oleate (400 $\mu M$) for 24 h. **(E)** Quantitation of CCTα-positive nLDs in oleate-treated cells. **(F, G)** The size distribution of cLDs (panel F) and nLDs (panel G) in oleate-treated cells was determined by cross-sectional area binning. **(H)** Average cross-sectional area of total and CCTα-positive nLDs was quantified in oleate-treated cells. In panels (C, D, E, F, G, H), results are presented as box and whisker plots showing the mean and 5th–95th percentile for analysis of 50–100 cells from three separate experiments. Significance was determined by two-tailed $t$ test compared with matched U2OS cell controls (**$P < 0.01$; ***$P < 0.001$).

structures that did not correspond to cLDs, and occasionally on LipidTox Red–positive nLDs (Fig 5B, a–c). In contrast, the DAG biosensor was strongly associated with cLDs in PML KO cells and not observed on nLDs (Fig 5C, d–f).

To better assess the distribution of DAG on nLDs, a nuclear-localized DAG biosensor (nGFP-DAG) was made by inserting a tandem NLS into GFP-C1(2)δ. When transiently expressed in oleate-treated U2OS cells, nGFP-DAG was detected on two types of structures:

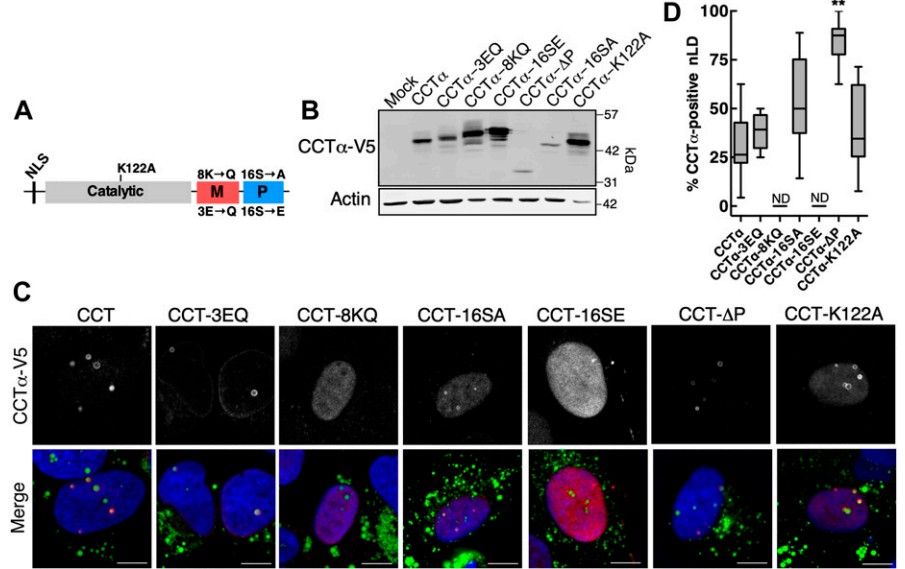

**Figure 3. Interaction of CCTα with nuclear lipid droplets (nLDs) is dependent on its M- and P-domains.**
**(A)** Domain organization of CCTα showing the location of catalytic, M-, and P-domain mutations. **(B)** U2OS cells were transiently transfected with empty vector (Mock) or the indicated CCTα–V5 constructs and treated with oleate (400 $\mu M$) for 24 h. Total cell lysates were immunoblotted with V5 and actin antibodies.
**(B, C)** Confocal images of U2OS cells (treated as described in panel B) immunostained with a V5 antibody. LDs and nuclei were visualized with BODIPY 493/503 and propidium iodide, respectively (bar, 10 $\mu m$).
**(D)** Quantification of the percentage of nLDs that were positive for wild-type and CCTα–V5 mutants presented as box and whisker plots showing the mean and 5th–95th percentile from analysis of 10–20 fields of cells from 2–3 separate experiments. Significance was determined by one-way ANOVA and Tukey's multiple comparison to CCTα–V5 (ND, not detected; **$P < 0.01$).

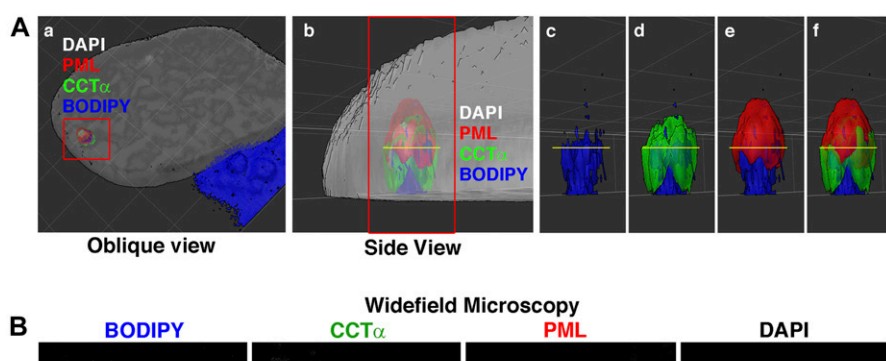

**Oblique view** **Side View**

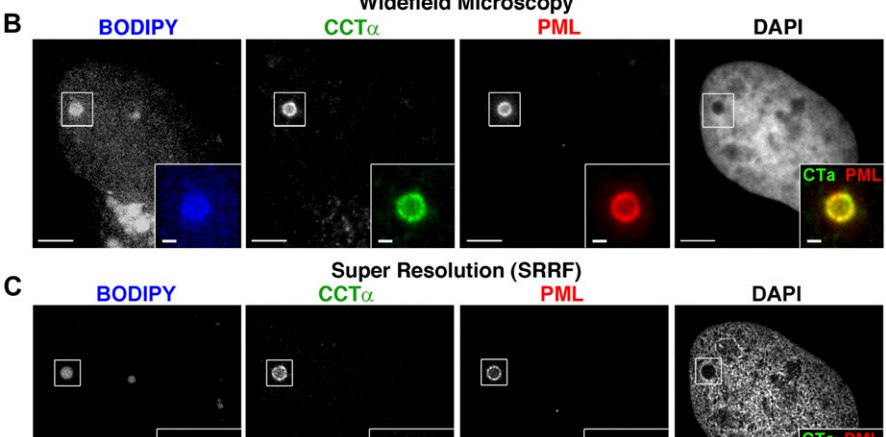

**Figure 4.** High-resolution imaging of lipid-associated PML structures and CCTα on nuclear lipid droplets (nLDs).

**(A)** 3D surface reconstruction of image stacks of oleate-treated (400 μM for 24 h) U2OS cells immunostained for PML and CCTα. LDs and nuclei were visualized with BODIPY 493/503 and DAPI, respectively. A large nLD was identified in association with CCTα and PML (panel a, red frame). The image was rotated and zoomed to produce a side view of the nLD (panel b, red frame). The framed structure in (b) was assessed by removing the DAPI channel (panels c–f) to reveal how the nLD/BODIPY (blue), CCTα (green), and PML (red) are associated. **(A, B)** Wide-field images of the same nLD in (A) imaged at the level of the yellow line indicated in panels b–f (image rotated ~90° clockwise). The nLD is highlighted and magnified in the inset (bar, 5 μm; inset scale bar, 1 μm). **(B, C)** Images of the same field of view in (B) were acquired by NanoJ SRRF. The nLD is highlighted and magnified in the inset (bar, 5 μm; inset bar, 1 μm).

small puncta that did not stain with LipidTox Red and PML-positive nLDs (Fig 6A). For the purpose of quantification, we divided LipidTox Red–positive nLDs in U2OS cells into four populations based on the presence or absence of DAG and/or PML (Fig 6B). This revealed that 60% of total nLDs were negative for both DAG and PML (DAG/PML–). Most of the remaining nLDs contained DAG and PML (79%), with the remainder containing only DAG (7%) or PML (14%) (Fig 6B, insert). The cross-sectional area of DAG/PML– nLDs was significantly reduced compared with total nLDs, whereas the PML+ and DAG/PML+ nLDs were significantly larger (Fig 6C). DAG+ nLDs were virtually absent from PML KO cells compared with total and DAG-negative nLDs, both of which were also significantly reduced (Fig 6D). We also observed that the nuclear sensor detected DAG on the abundant nLDs that form in oleate-treated Caco2 cells (Fig S8A). Collectively, the results in Figs 5 and 6 indicate that DAG is primarily enriched in large LAPS-positive nLDs, and that loss of these DAG-rich nLDs in PML KO leads to appearance of DAG in cLDs.

To determine whether DAG was a positive effector of CCTα association with nLDs, we determined whether CCTα was preferentially associated with DAG-positive nLDs. Immunofluorescence and line scans through nLDs in oleate-treated U2OS cells showed co-localization between the nuclear DAG sensor and CCTα (Fig 7A). However, in PML KO cells, there was evidence of CCTα-positive nLDs that lacked DAG, suggesting poor correlation between DAG content and CCTα. This conclusion was supported by quantitation of CCTα and DAG distribution on nLDs. Of the ~50% of nLDs in U2OS cells that contained DAG and/or CCTα, only 43% contained both DAG and CCTα, suggesting that the DAG content of an nLD is not a strong

indicator for recruitment of CCTα (Fig 7B, insert). Quantification of DAG/CCTα distribution on nLDs in PML KO cells indicated a significant reduction in all four populations compared with control cells, particularly those that contained DAG and DAG/CCTα, but the percent distribution of DAG and CCTα on nLDs was similar to control cells (Fig 7B, insert). As well, there was no significant difference in the cross-sectional area of nLDs containing DAG/CCTα in control versus PML KO cells (Fig 7C). The partial (30–40%) co-localization of CCTα and DAG on nLDs suggests DAG is not a critical factor in CCTα recruitment, whereas electrostatic interactions between lipids and charged residues in the CCTα M-domain could be the driving force for interaction with nLDs (Fig 3).

The source of DAG in nLDs could be Lipin1, a nuclear/cytoplasmic phosphatidic acid (PA) phosphatase that produces DAG for phospholipid and TAG synthesis (Csaki et al, 2013). The human and murine genes encode similar Lipin1α and β splice variants (Peterfy et al, 2005; Croce et al, 2007) that are partially localized to cLDs in COS cells and macrophages (Valdearcos et al, 2011; Wang et al, 2011), potentially recruited by seipin (Sim et al, 2012). To determine if Lipin1 recruitment to nLDs requires LAPS, transiently expressed Lipin1α and β were localized in U2OS and PML KO cells by immunofluorescence microscopy. V5-tagged Lipin1α was expressed in the nucleus and cytoplasm of untreated U2OS and PML KO cells (Fig 8A). Treatment of U2OS cells with oleate caused extensive association of Lipin1α with the surface of BODIPY-positive nLDs but not with cLDs (Fig 8B, a–c). Lipin1α was also localized to the surface of nLDs in Caco2 cells treated with oleate for 24 h (Fig S8B). To determine the co-localization of Lipin1α and its product DAG in U2OS

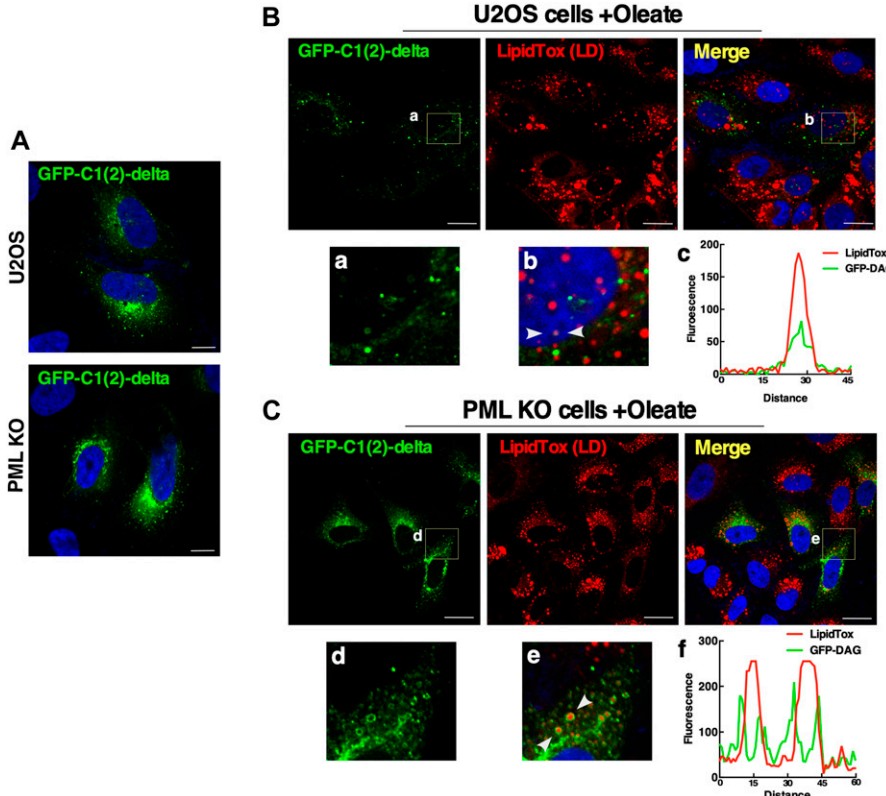

**Figure 5. Lipid-associated PML structures control DAG levels on cytoplasmic lipid droplets (cLDs) and nuclear LDs (nLDs).**
**(A)** Localization of the DAG sensor GFP-C1(2)δ in U2OS and PKL KO cells (bar, 10 μm). **(B)** U2OS cells expressing GFP-C1(2)δ were treated with 400 μm oleate for 24 h, and LDs were visualized with LipidTox Red (bar, 10 μm). Magnified areas from GFP and merged images are in panels (a) and (b). Arrows in panel (b) indicate the region selected for an RGB line scan plot (panel c) showing the association of the DAG sensor with a nLD. **(A, C)** PML KO cells were treated with oleate and stained as described in panel (A) (bar, 10 μm). Magnified areas from GFP and merge images are shown in panels (d) and (e). Arrows in panel (e) indicate the region selected for an RGB line scan plot (panel f) showing the association of the DAG sensor with cLDs.

cells, the distribution and cross-sectional area of Lipin1α- and DAG-containing nLDs was quantified. Of the DAG/Lipin1α-positive nLDs, 53% contained both (Fig 8C, insert), and Lipin1α-positive nLDs were significantly larger than other components of the population (Fig 8D). In oleate-treated PML KO cells, Lipin1α was diffusely localized in the cytoplasm and nucleus but was virtually absent from nLDs (Fig 8E), a conclusion that was confirmed by quantification of Lipin1α-positive nLDs in PML KO cells compared with controls (Fig 8F). The Lipin1β isoform also associated with the surface of nLDs but not cLD in oleate-treated U2OS cells (Fig S9A) and was not detected on nLDs in oleate-treated PML KO cells (Fig S9B). The Lipin1 substrate PA was detected with the GFP-nes-2xPABP (Bohdanowicz et al, 2013) biosensor on the plasma membrane and cytoplasmic membranes of control and PML KO cells but was absent from nLDs or cLDs (Fig S10). These data indicate that LAPS are required for association of Lipin1 with nLDs, which could account for the reduced DAG content of nLDs in PML KO cells (Fig 6).

## LAPS positively regulate PC and TAG synthesis

We next tested whether ablating LAPS in PML KO cells and reducing nLD-associated CCTα, DAG, and Lipin1 affected cellular lipid synthesis. PC biosynthesis was measured by [³H]choline pulse-labeling for 2 and 4 h in cells treated with or without oleate. PC synthesis in untreated PML KO cells was similar to controls (Fig 9A). However, PC synthesis was poorly activated in oleate-treated PML KO cells and reduced significantly compared with the 2.5-fold increase caused by oleate in U2OS cells (Fig 9A). PML KO had a minor inhibitory effect

on de novo synthesis of fatty acids from [³H]acetate that was only significant in cells cultured in lipoprotein-deficient serum (Fig 9B). TAG and CE synthesis in control and PML KO cells was measured by [³H]oleate incorporation (Fig 9C and D). Relative to controls, the incorporation of [³H]oleate into TAG and CE in PML KO cells cultured in FCS or lipoprotein-deficient serum was significantly decreased approximately twofold. To measure the de novo synthesis of TAG, control and PML KO cells were incubated with 100 μM oleate for up to 6 h in the presence of [³H]glycerol. In this case, incorporation into TAG in PML KO cells was increased at 1–4 h but returned to control values at 6 h (Fig 9E). [³H]Glycerol incorporation into TAG in U2OS and PML KO cells was similar after oleate treatment for 12 and 24 h (results not shown). Thus, despite accounting for only 10–15% of total cellular LDs (Fig 2), nLDs with associated CCTα and Lipin1 activities are important sites for the regulation of PC and TAG synthesis.

## Discussion

The INM is contiguous with the outer nuclear membrane and cytoplasmic ER (Ungricht & Kutay, 2017) and, thus, could receive lipids by lateral diffusion from the ER (van Meer et al, 2008). However, the INM in yeast (Romanauska & Kohler, 2018) and mammals (Peterson et al, 2011; Csaki et al, 2013; Aitchison et al, 2015; Haider et al, 2018) harbors lipid biosynthetic enzymes and regulatory proteins suggestive of compartmentalized nuclear lipid synthesis. Analogous to the ER, the INM is the site for de novo biogenesis of nLDs in yeast

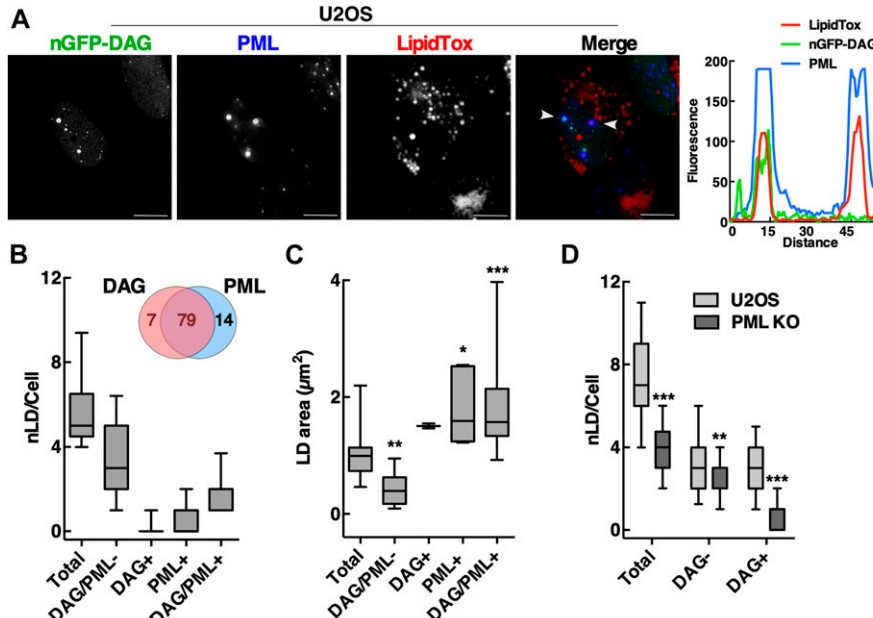

**Figure 6. LAP-positive nuclear lipid droplets (nLDs) are enriched in DAG.**
**(A)** U2OS cells transiently expressing nuclear localized GFP-C1(2)δ (nGFP-DAG) were exposed to oleate (400 µM) for 24 h and were fixed and immunostained with a PML antibody. LDs were visualized with LipidTox Red (bar, 10 µm). Arrows in the merged image indicate the region selected of an RGB line scan plot showing the localization of DAG-GFP with PML and nLDs. **(B)** Quantitation of nLDs in oleate-treated U2OS cells containing neither nGFP-DAG or PML (DAG/PML–), nGFP-DAG (DAG+), PML (PML+), or both (DAG/PML+). A Venn diagram shows the percent distribution of nGFP-DAG and PML-positive nLDs. **(C)** The average cross-sectional area of nLDs in oleate-treated U2OS cells containing nGFP-DAG and/or PML as described above. **(D)** Quantification of DAG-negative (–) and DAG-negative (+) nLDs in U2OS and PML KO cells. **(B, C, D)** Results are presented as box and whisker plots showing the mean and 5th–95th percentile from analysis of 50–100 cells in three separate experiments. Significance was determined by one-way ANOVA and Tukey's multiple comparison to total nLD area (panel C) or two-tailed *t* test compared with U2OS controls (panel D) (*$P < 0.05$; **$P < 0.01$; ***$P < 0.001$).

(Romanauska & Kohler, 2018), whereas in hepatoma cells, nLDs are derived from luminal ER LDs that enter the nucleus after dissolution of the INM (Soltysik et al, 2019). PML expression, in particular the PML-II isoform, aids in nLD biogenesis at the INM by disrupting lamin-free regions of the INM to allow luminal LD release (Ohsaki et al, 2016). Ohsaki et al (2016) also demonstrated that more than 70%

of PML in the nuclei of hepatoma cells were associated with the nLDs, but their function and relationship to PML NBs and other PML structures is unknown. We propose that these atypical PML structures be referred to as LAPS to differentiate them from canonical PML NBs and recognize their association with nLDs. LAPS are not essential for nLD formation in U2OS cells but are necessary

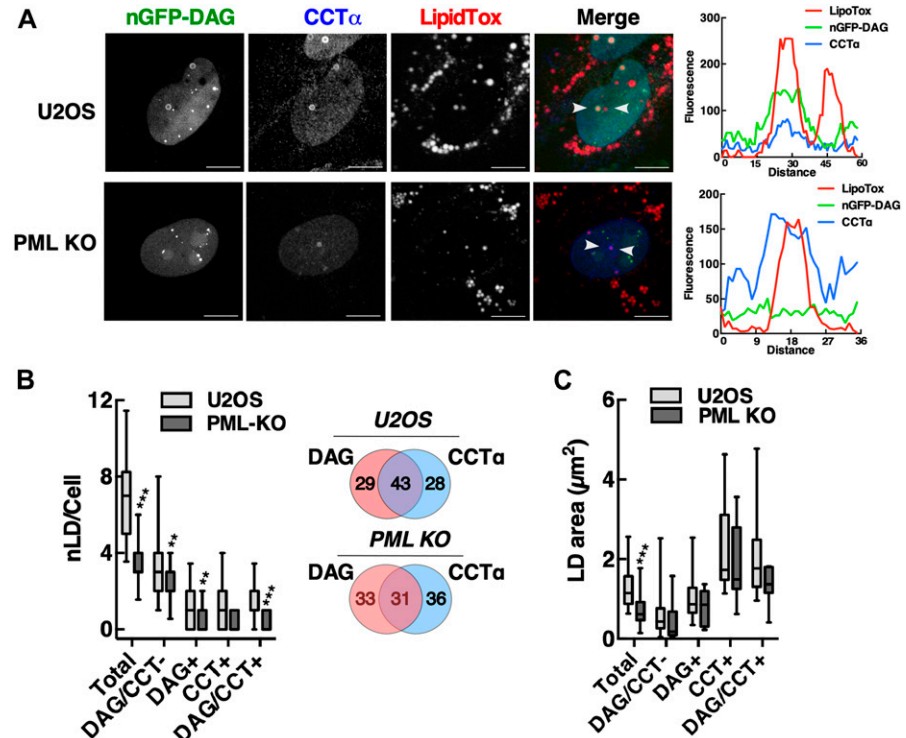

**Figure 7. CCTα association with nuclear lipid droplets (nLDs) is DAG independent.**
**(A)** U2OS and PML KO cells transiently expressing a nuclear nGFP-DAG sensor were exposed to 400 µM oleate for 24 h, fixed, and immunostained with a CCTα antibody and incubated with LipidTox Red to visualize LDs (bar, 10 µm). Arrows in the merged image indicate the region selected for an RGB line scan plot showing the association of CCTα and DAG-GFP. **(B)** Quantification of nLDs in oleate-treated U2OS and PML KO cells containing neither nGFP-DAG or CCTα (CCTα/DAG-GFP-), nGFP-DAG (DAG+), CCTα (CCT+), or both (CCT/DAG+). A Venn diagram shows the percent distribution of nGFP-DAG and CCTα-positive nLDs. **(C)** In oleate-treated U2OS and PML KO cells, the cross-sectional area of nLDs containing nGFP-DAG and/or CCTα was measured. **(B, C)** Results are presented as box and whisker plots showing the mean and 5th–95th percentile for analysis of 50–100 cells from three separate experiments. Significance was determined by a two-tailed *t* test compared with matched U2OS controls (**$P < 0.01$; ***$P < 0.001$).

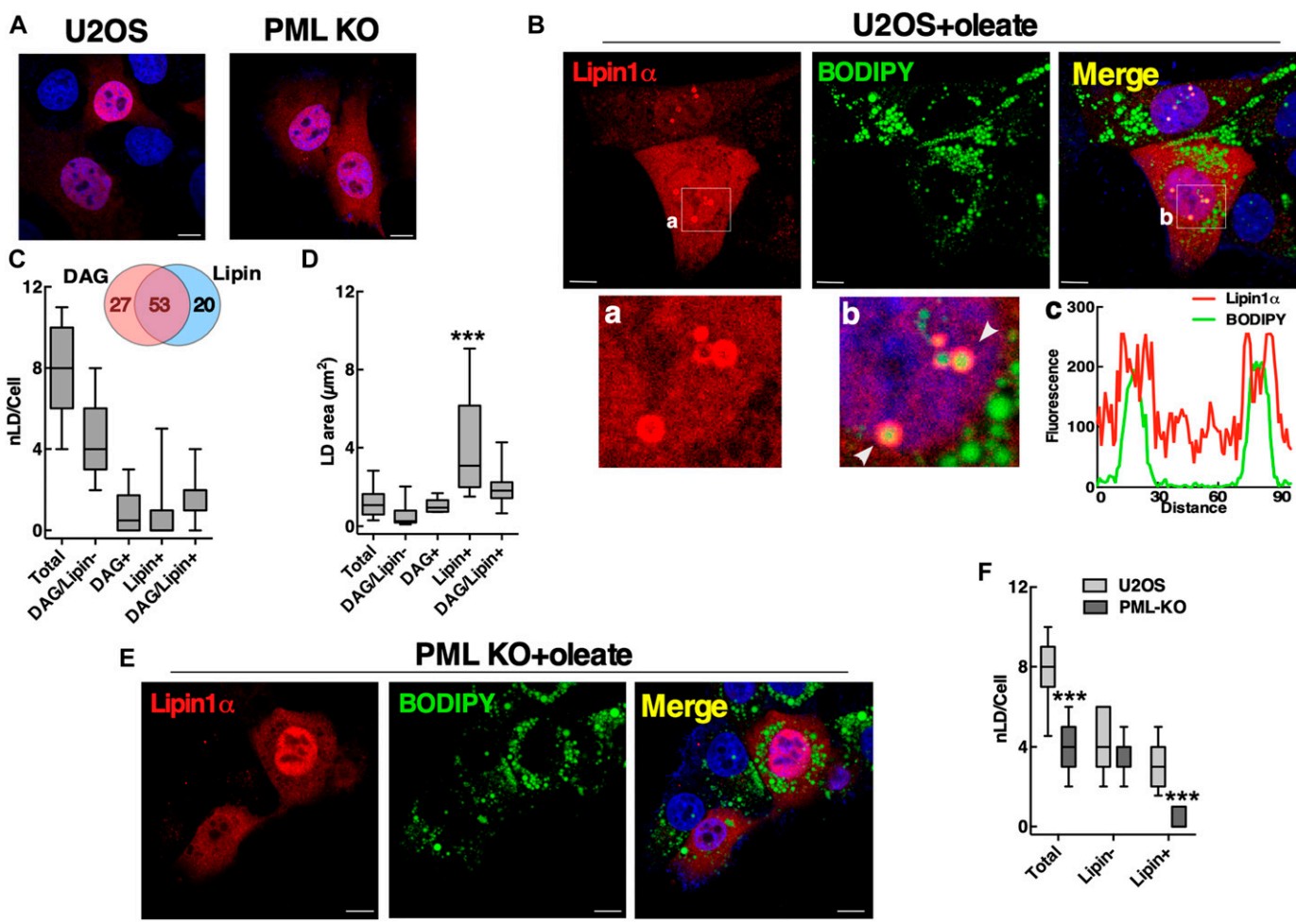

**Figure 8. Lipid-associated PML structures promote formation of Lipin1α with DAG-positive nuclear lipid droplets (nLDs).**
**(A)** U2OS and PML KO cells transiently expressing Lipin1α-V5 were immunostained with a V5 monoclonal antibody, and nuclei were visualized with DAPI (bar, 10 μm).
**(B)** U2OS cells transiently expressing Lipin1α-V5 were treated with 400 μM oleate for 24 h and immunostained with a V5 monoclonal antibody. LDs were visualized with BODIPY 493/503 (bar, 10 μm). Panels (a) and (b) show magnified regions for Lipin1α and merge images. Arrows show the region in panel (b) selected for an RGB line scan plot (panel c) showing Lipin1α on the surface of nLDs. **(C)** Quantification of nLDs in oleate-treated U2OS cells containing neither nGFP-DAG nor Lipin1α (Lipin/PML−), nGFP-DAG (DAG+), Lipin1α (Lipin+), or both (Lipin/DAG+). A Venn diagram shows the percent distribution of nGFP-DAG and Lipin1α-positive nLDs. **(D)** The cross-sectional area of nLDs containing differing compositions of nGFP-DAG, and Lipin1α was quantified in oleate-treated U2OS cells. **(E)** PML KO cells were treated with oleate and immunostained as described above (bar, 10 μm). **(F)** Quantification of Lipin-negative (−) and Lipin-positive (+) nLDs in U2OS and PML KO cells. **(C, D, F)** Results are presented as box and whisker plots showing the mean and 5th–95th percentile for analysis of 50–100 cells from three separate experiments. **(C, D, F)** Significance was determined by one-way ANOVA and Tukey's multiple comparison (panels C and D) or two-tailed t test compared with control U2OS cells (panel F) (***P < 0.001).

for the functional maturation of the nLD such that it can recruit CCTα and Lipin1 for the regulation of PC and TAG synthesis.

Coincident with the appearance of LAPS on nLDs in oleate-treated U2OS cells was the loss of canonical PML NBs containing SUMO1, DAXX, and SP100. Although oleate treatment caused PML proteins to transfer quantitatively to nLDs without affecting expression, the resultant LAPS were poorly SUMOylated and depleted of PML NB resident proteins. Electron microscopy showed the surface of nLDs had the typical morphology of PML NB (Ohsaki et al, 2016), indicating that LAPS are structurally related but have lost features that are required for PML NB assembly, notably SUMO1 and SP100. Importantly, because LAPS are deficient in these normally constitutive protein components of this nuclear subdomain, by convention and definition, they cannot be called PML NBs. By

making this distinction, LAPS are placed within the continuum of PML-containing structures that are already described in the literature with PML NBs at one extreme, followed by LAPS, and at the other extreme, mitotic accumulation of PML proteins (MAPPs) (Dellaire et al, 2006) and PML rods and rosettes in embryonic stem cells (Butler et al, 2009) that are completely devoid of DAXX, SP100, and SUMO1.

PML KO U2OS cells had a 40–50% reduction in the number of nLDs, a shift to smaller size and reduced association of CCTα, Lipin1, and DAG. However, large CCTα-positive nLDs in PML KO cells were only reduced by 50%, indicating other mechanisms for nLD biogenesis. Microsomal triglyceride transfer protein inhibitors did not affect nLD formation in U2OS cells suggesting luminal LDs may not be precursors (Soltysik et al, 2019). However, PML-negative nLDs

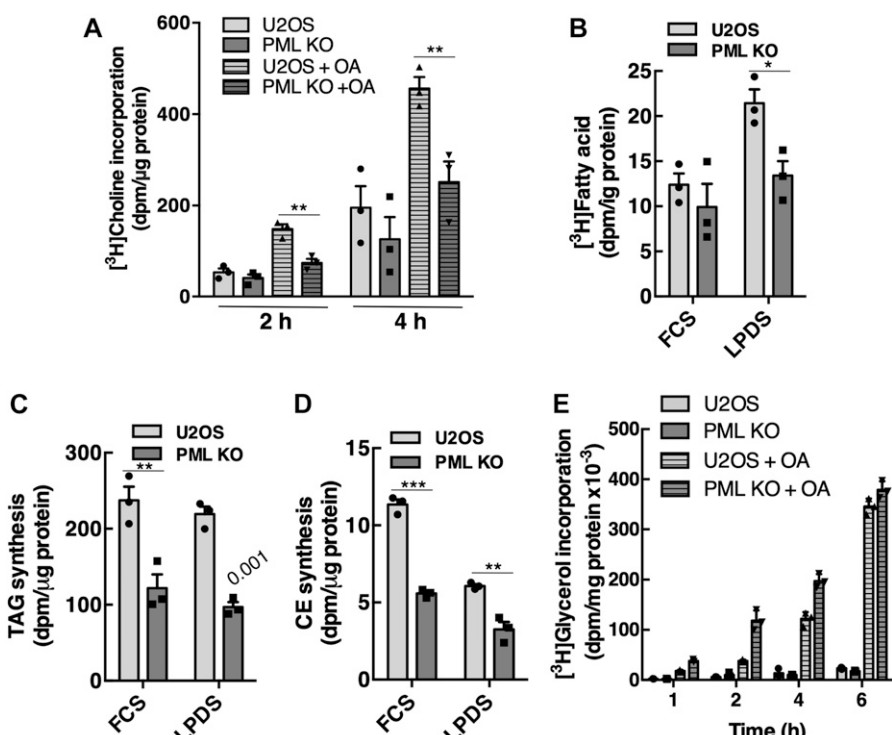

**Figure 9. PML KO cells have reduced PC and TAG synthesis.**
**(A)** Cells were cultured for 24 h in medium containing no addition or 300 µM oleate (OA), incubated in choline-free medium with [³H]choline (1 µCi/ml) for 2 or 4 h, and isotope incorporation into PC was quantified. Results are the mean and SD of five experiments. **(B)** U2OS and PML KO cells were cultured in FCS or lipoprotein-deficient serum (LPDS) for 24 h, incubated with [³H]acetate for 4 h, and incorporation into fatty acids was determined. Results are the mean and SD of three experiments. **(C, D)** U2OS and PML KO cells were cultured in FCS or LPDS for 18 h before incubation with 100 µM [³H]oleate for 4 h to measure TAG and cholesterol ester (CE) synthesis. Results are the mean and SD of three experiments. **(E)** U2OS and PML-KO cells were labelled with [³H]glycerol (2 µCi/ml) in the presence or absence of oleate (OA, 100 µM) for up to 6 h. [³H]Glycerol incorporation into TAG is the mean and SD of triplicate determinations from a representative experiment. Statistical comparisons by two-tailed $t$ test (*$P < 0.05$; **$P < 0.01$; ***$P < 0.0001$).

were observed in close proximity to the NE suggesting that nLDs could originate from ER luminal precursors and/or de novo lipid synthesis at the INM. A clue to the potential mechanisms for nLD biogenesis comes from high-resolution imaging of the polarized localization of LAPS on nLDs. PML-II patches on the INM have been proposed as sites where nLD emerge from the INM (Ohsaki et al, 2016). We used 3D imaging to reveal that these LAPS persist on small and large nLDs and could represent the surface at which the nascent nLD acquired PML proteins as it emerged from the INM. The subsequent maturation of the nLD could be facilitated by the recruitment and activation of CCTα and Lipin1 on exposed surfaces of the LD monolayer, resulting in increased PC and TAG synthesis. The emergence of nLDs from the INM could also involve the CCTα M-domain, which is capable of inducing positive membrane curvature and expansion of the NR (Lagace & Ridgway, 2005; Taneva et al, 2012). In this scenario, translocation of CCTα to the INM could induce NR formation and destabilize the bilayer to facilitate the release of nascent nLDs. The presence of large CCTα-positive nLDs in PML KO cells supports this concept.

Because PML KO reduced CCTα-positive nLDs, we conclude that LAPS are essential to activate PC synthesis in response to prolonged exposure to fatty acids by providing a lipid surface for CCTα activation. Analysis of CCTα mutants showed that association with nLDs was mediated by basic and acidic residues in the M-domain and by dephosphorylation of the P-domain. However, not all 16 phosphosites in the P-domain must be dephosphorylated to promote CCTα association with nLDs. Notably, nLD-associated CCTα is dephosphorylated at S319, whereas Y359 and S362 are constitutively phosphorylated (Yue et al, 2020). To identify lipids that promote CCTα interaction with nLDs, we initially focused on DAG, which

induces packing defects in membranes into which the CCTα M-domain can insert (Cornell & Ridgway, 2015; Cornell, 2016). PML expression was positively correlated with large nLDs that were enriched in DAG. However, DAG and CCTα were not preferentially co-localized in nLDs nor was their relative distribution in nLDs altered by loss of PML expression. Although the exclusion of CCTα and the biosensor on nLDs could be due to competition for DAG, it is more likely that factors such as surface curvature, enrichment in anionic lipids, or a high PE/PC ratio promote the association of CCTα. For instance, the high PE/PC ratio in insect cLDs relative to human cLDs (Jones et al, 1992; Tauchi-Sato et al, 2002) could be responsible for nuclear export and localization of CCT1 and CCTα on cLDs during oleate loading in *Drosophila* S2 cells (Krahmer et al, 2011).

The striking PML-dependent redistribution of DAG between nLDs and cLDs was linked to the nuclear PA phosphatase Lipin1α, which co-localized with DAG on nLDs in oleate-treated U2OS and Caco2 cells. Because small nLDs in PML KO cells were relatively devoid of both Lipin1α and DAG, Lipin1α appears to provide DAG for nuclear TAG synthesis by diacylglycerol acyltransferase 2, which was localized to nLDs when ectopically expressed in Huh7 cells (Ohsaki et al, 2016). In yeast, nLDs are enriched in PA and DAG as well as enzymes for neutral lipid and phospholipid synthesis (Romanauska & Kohler, 2018). Interestingly, the yeast Lipin homologue Pah1p was not detected on nLDs but was active on the INM where it produced DAG used for TAG incorporation into nLDs. In U2OS cells, the factor(s) mediating the selective association of Lipin1 with nLDs are uncertain. PA, the Lipin1 substrate that promotes its association with membranes (Ren et al, 2010), was not detected on nLDs or cLDs, suggesting it is below the biosensor detection threshold or the biosensor is not sufficiently PA-specific, as was suggested

previously (Horchani et al, 2014). Similarly, Lipin1α and PA were not detected on cLDs in U2OS or PML KO cells, and thus not directly implicated in the accumulation of DAG on cLDs in PML KO cells (Fig 5C).

Metabolic labeling experiments support the concept that LAPS and nLDs are sites for TAG synthesis and regulation. PML KO cells had significantly reduced [³H]oleate incorporation into TAG, reduced nLDs, and increased levels of DAG in cLDs, indicative of a block in nuclear and cytoplasmic TAG synthesis. However, PML KO did not inhibit de novo synthesis TAG from [³H]glycerol, suggesting a more complex role in oleate uptake and utilization. Prior studies have indicated complex, tissue-specific roles for PML in fatty acid metabolism. For example, PML NB–dependent activation of peroxisome proliferator-activated receptor (PPAR)γ co-activator-1α (PGC-1α), PPAR signaling and fatty acid β-oxidation were shown to provide a growth advantage to breast cancer cells (Carracedo et al, 2012), and controlled asymmetric division and maintenance of hematopoietic stem cells (Ito et al, 2012). In contrast, PML KO mice had tissue-specific enhancement of both fatty acid β-oxidation and synthesis, increased metabolic rate, and resistance to diet-induced obesity (Cheng et al, 2013). Together with these findings, our results indicate that nLDs and LAPS could have a multi-facetted role in lipid homeostasis involving the recruitment of CCTα and Lipin1 to promote PC and TAG synthesis and storage as well as Lipin1 regulation of a PGC-1α/PPAR signaling pathways that control fatty acid uptake, storage, and oxidation (Finck et al, 2006).

# Materials and Methods

## Cell culture

U2OS (ATCC HTB-96) and CaCo2 (ATCC HTB-37) cells were cultured in DMEM containing 10% FBS (medium A) at 37°C in a 5% $CO_2$ atmosphere. CRISPR/Cas9 methodology was used to knockout the expression of all PML isoforms in U2OS cells (Attwood et al, 2019). Unlabeled and [³H]oleate/BSA complexes (6:1, mol/mol, 12 mM oleate stock solutions) were prepared as described (Goldstein, 1983).

## Plasmid transfection

Cells were transfected with plasmids encoding murine Lipin1α-V5 and Lipin1β-V5 (Bou Khalil et al, 2009) (Zemin Yao, University of Ottawa), EGFP-PML isoforms (Bischof et al, 2002) (Oliver Bischof, Institute Pasteur), GFP-C1(2)δ (Tobias Meyer, plasmid #21216; Addgene), GFP-nes-2xPABP (Sergio Grinstein, University of Toronto), CCTα-16SA and CCTα-16SE (P-domain phosphorylation mutants) (Wang & Kent, 1995), and CCTα-3EQ and CCTα-8KQ (M-domain mutants) (Johnson et al, 2003; Gehrig et al, 2009) using Lipofectamine 2000 (2 μl/μg DNA) according to the manufacturer's instructions (Life Technologies). A nuclear DAG sensor (nGFP-DAG) was prepared by subcloning a ×2 NLS cassette into the BsrGI-AflII site of pGFP-C1(2)δ. Experiments were initiated 24–36 h after plasmid transfection.

## Immunoblotting

Cells were lysed in SDS buffer (12.5% SDS, 30 mm Tris–HCl, 12.5% glycerol, and 0.01% bromophenol blue [pH 6.8]) and heated at 95°C for 5 min. Lysates were separated by SDS–PAGE, transferred to nitrocellulose membranes, and incubated in TBS (20 mM Tris–HCl [pH 7.4] and 500 mM NaCl):Odyssey Blocking Buffer (5:1, vol/vol) for 1 h. Nitrocellulose was blotted using primary antibodies against human CCTα (Morton et al, 2013), PML (rabbit polyclonal A301–167A; Bethyl Laboratories), V5 monoclonal (MCA-1360; Bio-Rad), or β-actin (mouse monoclonal AC15; Sigma-Aldrich). Proteins were visualized with goat antimouse or goat antirabbit IRDye-800 or IRDye-680 secondary antibodies (LI-COR Biosciences) using an Odyssey Imaging System and application software (v3.0).

## Analysis of PC synthesis using [³H]choline incorporation

U2OS and PML KO cells were incubated in the presence or absence of 400 μM oleate/BSA for 24 h. Cells were rinsed twice with choline-free medium A and then incubated with choline-free medium A containing [³H]choline (2 μCi/ml) for 2 and 4 h. After isotope labeling, the cells were rinsed with cold PBS, harvested in methanol:water (5:4, vol/vol), [³H]PC was extracted in chloroform/methanol, and the radioactivity quantified by liquid scintillation counting and normalized to total cellular protein (Storey et al, 1997).

## Analysis of fatty acid and TAG synthesis

U2OS and PML KO cells were incubated with 2.5 μCi/ml [³H]acetate for 4 h to measure fatty acid synthesis (Brown et al, 1978). Briefly, lipid extracts from cells were saponified in ethanol and 50% potassium hydroxide (wt/vol) for 1 h at 60°C and extracted with hexane. The hydrolysate was then acidified with HCl (pH < 3), and radiolabeled fatty acids were extracted with hexane and quantified by liquid scintillation counting. Incorporation of [³H]acetate into fatty acids was normalized to total cellular protein.

TAG and CE synthesis was determined by incubating cells with 100 μM [³H]oleate/BSA. After 4 h, the cells were rinsed twice with cold 150 mM NaCl and 50 mM Tris–HCl (pH 7.4) with 2 mg/ml BSA and once with the same buffer without BSA. [³H]Oleate-labeled lipids were extracted from dishes with hexane:isopropanol (3:2, vol/vol). Samples were dried under nitrogen, resolved by thin-layer chromatography in hexane:diethyl ether:acetic acid (90:30:1, vol/vol), and radioactivity in [³H]TAG and [³H]CE was quantified by scintillation counting and normalized to total cellular protein. De novo TAG synthesis was determined by incubating cells with or without 400 μM oleate/BSA in the presence of 2 μCi/ml [³H]glycerol. The cells were rinsed with cold PBS, and radioactive lipids were extracted and quantified as described above.

## Immunofluorescence microscopy

Cells cultured on glass coverslips were fixed with 4% (wt/vol) paraformaldehyde and permeabilized for 10 min with 0.1% (wt/vol) Triton X-100 at room temperature. Coverslips were incubated with primary antibodies against CCTα, PML (monoclonal E-11; Santa Cruz), PML (rabbit polyclonal A301–167A; Bethyl Laboratories),

LMNA/C (monoclonal 4C11; Cell Signaling), emerin (polyclonal FL-254; Santa Cruz), DAXX (polyclonal D7810; Sigma-Aldrich), SP100 (polyclonal PA5-53602; Invitrogen), SUMO1 (rabbit monoclonal Y299, Ab32058; Abcam), or V5 in PBS containing 1% (wt/vol) BSA. This was followed by secondary AlexFluor488-, 594-, and 647-conjugated goat antirabbit or antimouse secondary antibodies (Thermo Fisher Scientific). To visualize nuclear and cLDs, BODIPY 493/503 or LipidTox Red was diluted 1:1,000 or 1:500, respectively, and incubated with the secondary antibodies. Coverslips were mounted on glass slides in Mowiol 4-88, and confocal imaging was performed using a Zeiss LSM510 or LSM710 laser scanning confocal microscope with a Plan-Apochromat 100× (1.4 NA) oil immersion objective. LD cross-sectional area in confocal images was quantified using ImageJ software (v1.47, National Institutes of Health). Images were converted to 8-bit, the threshold was adjusted, and the "analyze particle" command was used to exclude on edges. The percent distribution of LDs within a binned area group was quantified.

Quantification of SP100, SUMO1, and DAXX co-localization with PML on BODIPY-positive nLDs was binned into strong, weak, and none association groupings. Strong association was indicated by co-localization of SP100, DAXX, and SUMO1 throughout the PML structure with a signal intensity that was >50% of that for PML. Weak association was scored as partial co-localized with PML on nLDs and a signal intensity <50% relative to PML. None-association indicated no signal associated with PML-positive nLDs.

For SRRF imaging (Gustafsson et al, 2016), immunostained cells (described above) were observed using a 100 X Plan-Apochromat (1.46 NA) oil immersion objective lens (Zeiss) by wide-field imaging on a Marianis microscope (Intelligent Imaging Innovations, 3i) based on a Zeiss Axio Cell Observer equipped with LED-based illumination via a SPECTRA III Light Engine (Lumencor), and a Prime BSI back-illuminated scientific complementary metal–oxide–semiconductor (sCMOS) camera (Teledyne Photometrics). This imaging configuration resulted in an effective pixel size of 65 × 65 nm in the captured images. To generate super-resolution images, 100 wide-field images were captured at 10 ms/frame using SlideBook 6 software and then exported to ImageJ (version 1.52, NIH) in a 16-bit Open Microscopy Environment-Tagged Image File Format. Image processing used a custom SRRF algorithm (NanoJ-LiveSRRF, available on request from Ricardo Henriques, University College London/Francis Crick Institute, UK) with the following settings: radius 3, sensitivity 2, magnification 4, average temporal analysis, and with intensity weighting and both vibration and macro-pixel correction turned on. NanoJ-LiveSRRF is the newest implementation of NanoJ-SRRF within the ImageJ software, available upon request as above. However, NanoJ-SRRF is already freely available (Gustafsson et al, 2016).

For confocal imaging and 3D volume rendering, the cells were immunostained as above and imaged using a 100 X Plan-Apochromat (1.46 NA) oil immersion objective lens (Zeiss) by spinning-disk confocal microscopy on a Marianis microscope (Intelligent Imaging Innovations, 3i) based on a Zeiss Axio Cell Observer equipped with Yokagawa CSU-X1 spinning-disk unit, four laser lines (405, 488, 560 and 640 nm), and a Prime95B (Teledyne/Photometrics). 3D volume rendering of confocal image stacks was performed using SlideBook 6 software (3i).

# Supplementary Information

# Acknowledgements

We would like to thank Romain Laine and Ricardo Henriques (University College London/Francis Crick Institute, UK) for providing guidance on SRRF imaging and optimization. This work was funded by a Project Grant to G Dellaire and ND Ridgway from the Canadian Institutes of Health Research (PJT62390) and the Bernard and Winnifred Lockwood Endowment for Research. J Lee was supported by a scholarship from the Nova Scotia Health Research Foundation. G Dellaire and ND Ridgway are Senior Scientists of the Beatrice Hunter Cancer Research Institute.

## Author Contributions

J Lee: formal analysis, investigation, methodology, and writing—original draft, review, and editing.
J Salsman: formal analysis, investigation, methodology, and writing—original draft, review, and editing.
J Foster: formal analysis, investigation, methodology, and writing—review and editing.
G Dellaire: conceptualization, data curation, supervision, funding acquisition, methodology, project administration, and writing—original draft, review, and editing.
ND Ridgway: conceptualization, data curation, supervision, funding acquisition, methodology, project administration, and writing—original draft, review, and editing.

## Conflict of Interest Statement

The authors declare that they have no conflict of interest.

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
