## [Reviewer comments · Life Science Alliance]

Life Science Alliance

Lipid-Associated PML Structures assemble nuclear lipid droplets containing CCT α and Lipin1

Jonghwa Lee, Jayme Salsman, Jason Foster, Graham Dellaire, and Neale Ridgway

DOI: <https://doi.org/10.26508/lsa.202000751>

Corresponding author(s): Neale Ridgway, Dalhousie University and Graham Dellaire, Dalhousie University

Review Timeline:

Submission Date:	2020-04-22
Editorial Decision:	2020-04-23
Revision Received:	2020-05-13
Accepted:	2020-05-14

Scientific Editor: Andrea Leibfried

Transaction Report:

Please note that the manuscript was previously reviewed at another journal and the reports were taken into account in the decision-making process at Life Science Alliance.

Reviewer #1 Review

Comments to the Authors (Required):

In this manuscript, the authors aimed to further investigate the dynamics of nuclear LDs in U2OS cells. They proposed a LAP domain that specifically associate with PML-II structures which are also devoid of canonical PML NB proteins. They further show that knocking out PML reduced nuclear LDs, and that CCTa and PML occupy distinct regions of nLDs. DAG and lipin1 can associate with nLDs through LAP. While there may be some interesting data, the overall quality of this work is low, and there is no major advancement in knowledge from Oshaki et al, JCB 2016; Soltyzik et al., 2019. Major concerns are below:

1. This reviewer is confused by the LAP domain. Is it part of the nuclear LDs or it is part of the PML? What is the molecular composition of this domain?
2. In figure S1 of Soltyzik et al, the formation of nLDs of U2O2 cells is independent of MTP. Thus, these nLDs are not those nLDs in hepatocytes. How are nLDs formed in U2OS cells? This is the most important question the authors should address.

3. Another critical question is the type of nLDs in U2OS cells: are they in the lumen or in the nucleoplasm? TEM needs to be employed to address this. The authors should refer to Soltysik et al on how to address these questions with appropriate techniques. For instance, using Lamins to separate nucleoplasmic and luminal LDs.

4. Immuno EM would help figure 4B.

5. It would be good to verify the DAG sensor with positive and negative controls.

6. The authors quoted a paper suggesting that lipin1 may localize to LDs through seipin. Since seipin is a key protein in LD biogenesis, its role in nLD formation in this particular cell line should be examined.

Reviewer #2 Review

Comments to the Authors (Required):

In this exhaustive study that primarily relies on microscopic experiments with manipulated cell types the authors demonstrate the existence of so-called LAP domains within the nuclear lipid droplet. Additionally, these structures contain DAG as well as the phosphocholine cytidyltransferase (CCT) and PA phosphatase (lipin 1) enzymes. The images and the analysis of them are of very high quality and provide novel information that advances understanding of nuclear lipid droplet biology. The analysis of lipid synthesis, which utilizes appropriate radioactive lipid precursors, is well done substantiating the conclusion that the LAP domain impacts on lipid synthesis. Yet, the inability to detect the PA phosphatase substrate PA in the LAP domain sheds some doubt on the model. Appropriately, the authors are up front with this negative observation allowing the reader to make their own conclusions.

One conclusion that this reader is uncomfortable with is that the LAP domain regulates lipid synthesis by recruitment of the enzymes. At this point, there are correlations that are suggestive of a regulatory mechanism, but the mechanism is not really clear at this point in the investigation. Although the manuscript is well written, the verbiage is a little verbose and speculative. The graphs are well constructed, but the presentation is drab (grey) when compared to the colorful images. Adding color to the bar charts and box plots is suggested.

Reviewer #3 Review

Comments to the Authors (Required):

In their study "Lipid-associated PML domains regulate CCT α , Lipin1 and lipid homeostasis" Lee et al. investigate the composition of PML domains associated with nuclear lipid droplets (nLDs) and the effects of a PML knockout on phosphatidylcholine and DAG metabolism. The mainly microscopy-based study in U2OS cells adds some insights into the specific composition of the nLD associated PML bodies and gives some evidence that the PML binding to nLDs has functional relevance for lipid metabolism. However, the amount of new insights gained by this study is limited. Most findings of this study have already been published by other groups. Ohsaki et al., 2016 has already shown

that nuclear LDs associate with PML domains and that a PML knockdown reduces the number of nLDs. In addition, the activation of CCT by nLD binding and phosphatidylcholine synthesis has previously been reported (Sołtysik et al., 2019).

In my opinion, this study does not significantly advance our understanding of the role of nLDs in lipid metabolism and the function of the nuclear-cytosolic compartmentalization of lipid storage. The study remains on a descriptive level and does not provide insights into the cellular mechanism how LD-associated PML bodies influence lipid metabolism. Many crucial questions remain unanswered: How does PML target a subset of nLDs? How does PML effect targeting of CCT, lipin to nLDs. Is the influence of the PML knockdown on lipid metabolism direct or secondary?

The study would strongly profit from a more unbiased systematic analysis of the impact of PML on cellular lipid metabolism. The quality of the study and the power of the data would strongly increase if results were supported by complementary biochemical approaches and methods. In addition, the structure of the manuscript and the writing need editing. It is often hard to follow the conclusions and the logic flow is in some parts missing.

More focus should be set on the physiological relevance of the findings. In the discussion, the authors mention the interesting aspect that PML has been described to control lipid metabolism and expression of genes for energy metabolism by controlling PPAR activity and AMPK signaling in vivo. Indeed, a crucial question is, whether nLD localization of PML is required for PPAR activation. The authors completely neglect this aspect in the design of their experiments. The effects of PML localization on nLDs on transcriptional activity, expression levels of lipid synthesis enzymes and proteins involved in lipolysis, beta-oxidation or lipophagy are not addressed at all.

Major points:

- 1) How does PML bind to LDs? Does PML-II contain a hairpin structure, amphipathic alpha helix or any other domain that mediates the LD targeting? Is this domain only present on PML-II and not the other isoforms? The identification of the nLD binding domain would increase the mechanistic insights. In addition, Please provide an alignment of the PML isoforms in the supplemental material.
- 2) Does depletion of the NLS in PML-II abolish the specificity for nLDs? If Δ NLS-PML-II localizes to cLDs, does this mediate the localization of CCT, lipin to cLDs? Can Δ NLS-PML-II rescue the KO-phenotype on LD number and size?
- 3) PML KO reduces CCT targeting, how does it influence targeting of other LD proteins, Plin2, Rab18?
- 4) How does the interactome of PML change after oleate loading and nLD targeting? This might give information about the function of the PML on nLDs. How is the association to transcription factors known to interact with PML affected (e.g. FOXO1)?
- 5) How does the PPAR activation after oleate treatment change in PML KOs compared to wt cells? What happens to PPAR activation when the lipid droplet targeting domain of PML-II is deleted?
- 6) How does the PML-KO influence the levels of enzymes involved in PC, DAG, TG synthesis before and after oleate treatment?
- 7) The PML-KO influences TG and PC synthesis, but to which extent are total DAG, TG, PC, PE levels before and after oleate treatment affected? How does the PML-KO specifically change lipid levels in the nuclear or cytosolic fraction? This could be quantified e.g. by TLC, enzymatic assays or lipidomics.
- 8) Please provide a quantification in Figure 1: Which percentage of cellular PML co-localizes with SUMO with and without oleate? The classification of strong and weak associations seems somewhat random.
- 9) PML and CCT have little overlap on nLDs. How is the overlap with other LD proteins such as Lipin or Plin2? Is there a competition for binding sites?
- 10) Figure 5 needs quantification. Which percentage of cLDs and nLDs are positive for the DAG-reporter construct?

11) Are CCT and lipin recruited to the same subset of nLDs?

12) Do lipin and DGAT2, GPAT co-localize on nLDs?

Minor points

- The authors state that the U2OS cells are a good model for the investigation of nLDs. How abundant are the nLDs in U2OS cells? How is the ratio of cLD/nLDs?
- The title and abstract need major editing. The title should reflect a broader interest. The abstract should make the finding of this study more clear and set it into the context of previous findings.
- The definition of LAP domain is slightly misleading. It should be clarified that it is used to describe a subset of PML bodies and not a protein domain.
- The last paragraph of the introduction needs editing. How can CRISPR KO identify a domain or determine the composition? The context and logic are missing.
- The presentation of the microscopy data should be improved with showing inlays with higher magnification in all figures.

April 23, 2020

RE: Life Science Alliance Manuscript #LSA-2020-00751-T

Dr. Neale D Ridgway
Dalhousie University
Pediatrics and Biochemistry & Molecular Biology
Rm C306, CRC Bldg. Atlantic Research Centre, Dalhousie University, 5849 University Ave.
Halifax, Nova Scotia B3H 4R2
Canada

Dear Dr. Ridgway,

Thank you for transferring your manuscript entitled "Lipid-associated PML domains regulate CCT α , Lipin1 and lipid homeostasis" to Life Science Alliance. Your manuscript was peer-reviewed at another journal before, and the editors transferred those reports to us with your permission.

The reviewers who evaluated your study elsewhere appreciated the aim of your study and found it solid, but would have expected a further reaching conceptual advance and more in-depth insight. This does not preclude publication here, and we would thus like to invite you to submit a revised version of your manuscript to us based on the reviewer reports already at hand.

Please provide a full point-by-point response, and include the requested quantifications and text changes in the revised manuscript accordingly. Furthermore, please:

- provide the manuscript file in docx format
- Upload all figures, including supplementary figures as individual files; all figure legends (incl. suppl. Figure legends) should get added to the main manuscript file
- Add a callout in the manuscript text for Fig 2F (you mention 1F instead) and Fig 9C-E
- you mention fig. S7A1, C5, C6, but it is unclear what you are referring to, please fix
- you currently mention Fig 7D-F, but these panels do not exist
- the figure legend to Fig. 5 is incomplete (panel C not described)
- fill in all mandatory fields in our submission system when submitting the revised version
- link your ORCID iD to your profile in our submission system, you should have received an email with instructions on how to do so
- I realize that you have to re-work some figures to address the reviewer concerns; please keep in mind the following:
 - make sure that insets and magnifications match (eg. not matching well in Fig 5)
 - add scale bars where missing (eg. Fig. 2B, Fig. 5A-C, Fig. 6A, Fig. 7A, Fig. 8A,B,E, Fig. S1, Fig. S2, Fig. S5A,B, Fig. S8A,B, Fig. S9A,B, Fig. S10)

To avoid unnecessary delays in the acceptance and publication of your paper, please read the

following information carefully.

A. FINAL FILES:

B. MANUSCRIPT ORGANIZATION AND FORMATTING:

Sincerely,

Thank you for accepting the transfer of our manuscript to *Life Science Alliance*, and to the reviewers for offering insightful comments and suggestions for improvement. As per your instructions, we have revised the manuscript to address reviewer comments as well as those from the editorial office. Below is a point-by-point response to these comments indicating where changes and corrections were made to the manuscript and figures.

Reviewer #1 Comments

In this manuscript, the authors aimed to further investigate the dynamics of nuclear LDs in U2OS cells. They proposed a LAP domain that specifically associate with PML-II structures which are also devoid of canonical PML NB proteins. They further show that knocking out PML reduced nuclear LDs, and that CCT α and PML occupy distinct regions of nLDs. DAG and lipin1 can associate with nLDs through LAP. While there may be some interesting data, the overall quality of this work is low, and there is no major advancement in knowledge from Oshaki et al, JCB 2016; Sotysik et al., 2019. Major concerns are below:

We disagree with this last point. Our study describes several important findings that were not in the two papers from the Fujimoto lab. 1) We show that PML structures on nLDs are not typical PML NBs and are a new class that we call Lipid-Associated PML Structures (LAPS). 2) We show that PML KO leads to loss of functional nLDs. PML KO cells have nLDs (as was reported in the two papers mentioned above) but these residual nLDs do not have associated Lipin1 or DAG and have reduced CCT α content. 3) We define the mechanism of association of CCT α with nLDs involving its M- and P-domains and provide high resolution images that define the structure of nLDs in relation to CCT α and PML coverage. 4) this is the first study to show that Lipin1 associates with nLDs in the nucleus and is a probable source of DAG for TAG synthesis.

1. This reviewer is confused by the LAP domain. Is it part of the nuclear LDs or it is part of the PML? What is the molecular composition of this domain?

As mentioned above, we have dropped the 'domain' designation that seemed to cause confusion and now call them Lipid-associated PML structures (LAPS). These are a unique PML subdomain that differ from PML NBs due to association with lipids (ie. nLDs), requirement for PML-II and relative lack of canonical PML-NB proteins.

2. In figure S1 of Sotysik et al, the formation of nLDs of U2O2 cells is independent of MTP. Thus, these nLDs are not those nLDs in hepatocytes. How are nLDs formed in U2OS cells? This is the most important question the authors should address.

We agree with the reviewer that nLD formation in U2OS cells likely occurs by an MTP-independent mechanism. This was mentioned on page 15. Although nLDs could form in U2OS cell by a different mechanism, they are compositionally similar to hepatocyte and Caco2 cell nLDs that contain PML-II, CCT α , Lipin1 and DAG.

3. Another critical question is the type of nLDs in U2OS cells: are they in the lumen or in the nucleoplasm? TEM needs to be employed to address this. The authors should refer to Sotysik et al on how to address these questions with appropriate techniques. For instance, using Lamins to separate nucleoplasmic and luminal LDs.

Collectively, we estimate that >70% of nLDs were positive for PML and/or CCT α indicating they were present wholly or partially in the nucleoplasm. There is a population of nLDs that are negative for these nucleoplasmic markers on nLDs that probably represent smaller nascent LDs or those in the NE lumen. We used immunostaining for emerin, a protein on the inner nuclear membrane that localizes on nuclear membrane invaginations, to show that most nLDs and those that are PML-positive, are not associated with emerin-positive inner nuclear envelope (SFigure 2).

4. Immuno EM would help figure 4B.

We agree, however the immunofluorescence 3D reconstructions in Fig. 4 and SFig. 7 also show the distribution of CCT α and PML on nLDs at an appropriate resolution to interpret the relative positions of these nLD-associated proteins.

5. It would be good to verify the DAG sensor with positive and negative controls.

The DAG sensor has been validated in terms of its translocation to membranes in response to altered DAG generation (see Codazzi 2001). In the future we would like to demonstrate changes in DAG mass in nLDs and correlation this with metabolic production and consumption of DAG. We have included a reference to the Romanauska, A., and A. Kohler paper, which showed that DAG is detected on the INM and nLD (using a C1 DAG biosensor) in yeast by the action of Pah1, which supports our results here (page 17).

6. The authors quoted a paper suggesting that lipin1 may localize to LDs through seipin. Since seipin is a key protein in LD biogenesis, its role in nLD formation in this particular cell line should be examined.

Seipin associates with cLDs in mammalian cells (as indicated on bottom page 11). In yeast, Pah1p (Lipin homologue) and nLD biogenesis from the INM is regulated in part by seipin (Sei1). Future work could determine if mammalian seipin has a similar role in nLD formation. However, there is no evidence at present that human seipin is present in the nucleus .

Reviewer #2 Comments

In this exhaustive study that primarily relies on microscopic experiments with manipulated cell types the authors demonstrate the existence of so-called LAP domains within the nuclear lipid droplet. Additionally, these structures contain DAG as well as the phosphocholine cytidyltransferase (CCT) and PA phosphatase (lipin 1) enzymes. The images and the analysis of them are of very high quality and provide novel information that advances understanding of nuclear lipid droplet biology. The analysis of lipid synthesis, which utilizes appropriate radioactive lipid precursors, is well done substantiating the conclusion that the LAP domain impacts on lipid synthesis. Yet, the inability to detect the PA phosphatase substrate PA in the LAP domain sheds some doubt on the model. Appropriately, the authors are up front with this negative observation allowing the reader to make their own conclusions.

One conclusion that this reader is uncomfortable with is that the LAP domain regulates lipid synthesis by recruitment of the enzymes. At this point, there are correlations that are suggestive of a regulatory mechanism, but the mechanism is not really clear at this point in the investigation.

We have replaced this language to indicate that LAPS are required to form functional nLDs that can recruit CCT α and Lipin1 to their surface. This is the case with CCT α , which we know interacts via its lipid binding M-domain (Fig. 3).

Although the manuscript is well written, the verbiage is a little verbose and speculative. The graphs are well constructed, but the presentation is drab (grey) when compared to the colorful images. Adding color to the bar charts and box plots is suggested.

We have attempted to tighten up the writing and reduce speculative passages.

Reviewer #3 Comments

In their study "Lipid-associated PML domains regulate CCT α , Lipin1 and lipid homeostasis" Lee et al. investigate the composition of PML domains associated with nuclear lipid droplets (nLDs) and the effects of a PML knockout on phosphatidylcholine and DAG metabolism. The mainly microscopy-based study in U2OS cells adds some insights into the specific composition of the nLD associated PML bodies and gives some evidence that the PML binding to nLDs has functional relevance for lipid metabolism. However, the amount of new insights gained by this study is limited. Most findings of this study have already been published by other groups. Ohsaki et al., 2016 has already shown that nuclear LDs associate with PML domains and that a PML knockdown reduces the number of nLDs. In addition, the activation of CCT by nLD binding and phosphatidylcholine synthesis has previously been reported (Softysik et al., 2019). In my opinion, this study does not significantly advance our understanding of the role of nLDs in lipid metabolism and the function of the nuclear-cytosolic compartmentalization of lipid storage. The study remains on a descriptive levels and does not provide insights into the cellular mechanism how LD associated PML bodies influence lipid metabolism. Many crucial questions remain unanswered: How does PML target a subset of nLDs? How does PML effect targeting of CCT, lipin to nLDs. Is the influence of the PML knockdown on lipid metabolism direct or secondary?

The study would strongly profit from a more unbiased systematic analysis of the impact of PML on cellular lipid metabolism. The quality of the study and the power of the data would strongly increase if results were supported by complementary biochemical approaches and methods. In addition, the structure of the manuscript and the writing need editing. It is often had to follow the conclusions and the logic flow is in some parts missing.

More focus should be set on the physiological relevance of the findings. In the discussion, the authors mention the interesting aspect that PML has been described to control lipid metabolism and expression of genes for energy metabolism by controlling PPAR activity and AMPK signaling in vivo. Indeed, a crucial question is, whether nLD localization of PML is required for PPAR activation. The authors completely neglect this aspect in the design of their experiments. The effects of PML localization on nLDs on transcriptional activity, expression levels of lipid synthesis enzymes and proteins involved in lipolysis, beta-oxidation or lipophagy are not addressed at all.

Major points:

1) How does PML bind to LDs? Does PML-II contain a hairpin structure, amphipathic alpha helix or any other domain that mediates the LD targeting? Is this domain only present on PML-II and not the other isoforms? The identification of the nLD binding domain would increase the mechanistic insights. In addition, Please provide an alignment of the PML isoforms in the supplemental material.

Our data on PML-II is in the supplemental data and was meant to confirm the role of PML-II in U2OS cells. Ohsaki et al. previously showed that the nuclear periphery binding motif in the C-terminus of PML-II was involved in nLD localization. That paper also shows PML alignments and regions potentially involved in nLD attachment. Thus, including that information here would be repetitive.

2) Does depletion of the NLS in PML-II abolish the specificity for nLDs? If Δ NLS-PML-II localizes to cLDs, does this mediate the localization of CCT, lipin to cLDs? Can Δ NLS-PML-II rescue the KO-phenotype on LD number and size?

These are all excellent suggestions and important areas of future experimentation.

3) PML KO reduces CCT targeting, how does it influence targeting of other LD proteins, Plin2, Rab18?
These would be important areas of future experimentation.

4) How does the interactome of PML change after oleate loading and nLD targeting? This might give information about the function of the PML on nLDs. How is the association to transcription factors known to interact with PML affected (e.g. FOXO1)?

We show that the known PML-NB interacting proteins DAXX, Sp100 and SUMO1 are all decreased on nLDs in PML KO cells. A more extensive interactome analysis for LAPS in oleate loaded cells is underway.

5) How does the PPAR activation after oleate treatment change in PML KOs compared to wt cells? What happens to PPAR activation when the lipid droplet targeting domain of PML-II is deleted? ?

These are excellent suggestions that will be investigated.

6) How does the PML-KO influence the levels of enzymes involved in PC, DAG, TG synthesis before and after oleate treatment?

Preliminary experiments suggest Lipin1 expression is not affected by PML KO. These are areas that we will investigate in the future.

7) The PML-KO influences TG and PC synthesis, but to which extent are total DAG, TG, PC, PE levels before and after oleate treatment affected? How does the PML-KO specifically change lipid levels in the nuclear or cytosolic fraction? This could be quantified e.g. by TLC, enzymatic assays or lipidomic.

A preliminary lipidomic analysis indicated that PC mass were unaffected by PML KO. This is in agreement with results in other studies showing that changes in the rate of PC synthesis are counteracted by PC degradation such that total PC mass is unaffected. We plan a more in-depth analysis of the effect of PML KO on lipid homeostasis by isotopic methods and mass measurements.

8) Please provide a quantification in Figure1: Which percentage of cellular PML co-localizes with SUMO with and without oleate?

If I understand the question correctly, you would like to know how whether SUMOylation of total cellular PML is affected by oleate treatment. The relative intensity of SUMO1 staining in the nucleus did not change that much with oleate treatment but this reflects SUMOylation of all nuclear proteins not just PML. As well, not all PML is associated with LAPS and PML NBs (ie. diffuse nucleoplasmic) so it was difficult to make this quantification. The analysis was designed to show whether LAPS differ in the content of PML NB associated proteins. SUMO1 is an essential and constitutive component of PML NBs and SUMO1 co-localizes with almost 100% of PML NBs in untreated cells consistent with its essential role in PML NB assembly (page 6; Zhong et al.). The quantification in oleate treated cells suggests that PML NBs could be initially recruited to nLDs, where they are remodelled into non-canonical LAPS lacking the classic PML NB components SUMO1, DAXX and SP100.

The classification of strong and weak associations seems somewhat random.

We have clarified how co-localization was quantified into the 3 bins (complete, partial and non) based on signal intensity relative to PML and overlap on nLDs (see page 22).

9) PML and CCT have little overlap on nLDs. How is the overlap with other LD proteins such as Lipin or

Plin2? Is there a competition for binding sites? We have not directly tested this, but experiments are ongoing.

This will be examined in future studies.

10) Figure 5 needs quantification. Which percentage of cLDs and nLDs are positive for the DAG-reporter construct?

This is an interesting question that we can only address in qualitative terms. Fig. 5 shows results with the original GFP-PKC-C1-delta probe that is primarily cytoplasmic. Based on Fig. 5B (U2OS cells), this probe does not associate with cLDs but the small fraction of probe that enters the nucleus was identified on some nLDs. As shown in Fig. 5C (PML KO cells), the probe was on cLDs but we did not know whether its absence from nLDs is due to sequestration in the cytoplasm. Therefore, because of unequal distribution of the GFP-PKC-C1-delta probe between the nucleus and cytoplasm, it was not possible to accurately assess percent distribution on nLDs and cLDs. We made a nuclear-localized GFP-PKC-C1-delta probe (nGFP-DAG, Fig. 6) to demonstrate its presence on nLDs in control cells and almost complete absence in PML KO cells. We have more clearly summarized the distribution of DAG in nLD and cLD on page 10.

11) Are CCT and lipin recruited to the same subset of nLDs?

We attempted to address this question but encountered some technical difficulties when trying to perform these experiments, which required the use of GFP/Cherry-tagged PML-II and CCT α that gave poor results due to overexpression/transfection issues.

12) Do lipin and DGAT2, GPAT co-localize on nLDs?

We would like to address the localization of endogenous DGAT2 and GPATS to nLDs in future studies.

Minor points

- The authors state that the U2OS cells are a good model for the investigation of nLDs. How abundant are the nLDs in U2OS cells? How is the ratio of cLD/nLDs?

Based on data in Fig. 2C and D, there is on average about 4-6 nLDs/cell and nLDs account for about 10% of total LDs. This ratio is similar to hepatocytes.

- The title and abstract need major editing. The title should reflect a broader interest. The abstract should make the finding of this study more clear and set it into the context of previous findings.

We have attempted to edit the title and abstract for clarity and interest.

- The definition of LAP domain is slightly misleading. It should be clarified that it is used to describe a subset of PML bodies and not a protein domain.

To resolve this, we have substituted 'structures' for 'domain' and now call them LAPS to define their unique localization on nLDs and the fact that their structure and composition is still uncertain. We describe where LAPS fit in the spectrum of PML containing structures (based on their DAXX, SP100 and SUMO1 composition) on page 14-15.

- The last paragraph of the introduction needs editing. How can CRISPR KO identify a domain or determine the composition? The context and logic are missing.

The final paragraph has been rewritten for clarity and context.

- The presentation of the microscopy data should be improved with showing inlays with higher magnification in all figures.

Inlays with higher magnification are shown in images with multi-cell fields (Fig. 2 and Supplemental Fig. S1).

Additional corrections specified by the editor:

Please provide a full point-by-point response, and include the requested quantifications and text changes in the revised manuscript accordingly. Furthermore, please:

- Add a callout in the manuscript text for Fig 2F (you mention 1F instead) and Fig 9C-E -**This was mislabelled and is now corrected**
- you mention fig. S7A1, C5, C6, but it is unclear what you are referring to, please fix. **The objects referred to in S7A and C is now indicated more clearly.**
- you currently mention Fig 7D-F, but these panels do not exist –**This was an error in Fig 9 and is now corrected.**
- the figure legend to Fig. 5 is incomplete (panel C not described). **Panel C is now described in more detail.**
- fill in all mandatory fields in our submission system when submitting the revised version--**Done**
- link your ORCID iD to your profile in our submission system, you should have received an email with instructions on how to do so-**Done**
- I realize that you have to re-work some figures to address the reviewer concerns; please keep in mind the following:
 - make sure that insets and magnifications match (eg. not matching well in Fig 5). **This has been corrected by re-designating the magnified panels.**
 - add scale bars where missing (eg. Fig. 2B, Fig. 5A-C, Fig. 6A, Fig. 7A, Fig. 8A,B,E, Fig. S1, Fig. S2, Fig. S5A,B, Fig. S8A,B, Fig. S9A,B, Fig. S10)- **These have been added as suggested.**

We hope these responses satisfy the reviewer's concerns such that the manuscript is now acceptable for publication in LSA.

May 14, 2020

RE: Life Science Alliance Manuscript #LSA-2020-00751-TR

Dr. Neale D Ridgway
Dalhousie University
Pediatrics and Biochemistry & Molecular Biology
Rm C306, CRC Bldg. Atlantic Research Centre, Dalhousie University, 5849 University Ave.
Halifax, Nova Scotia B3H 4R2
Canada

Dear Dr. Ridgway,

Thank you for submitting your Research Article entitled "Lipid-Associated PML Structures assemble nuclear lipid droplets containing CCT α and Lipin1". I appreciate the response you provided to the reviewer concerns and the introduced changes, and it is a pleasure to let you know that your manuscript is now accepted for publication in Life Science Alliance. Congratulations on this interesting work.

DISTRIBUTION OF MATERIALS:

Again, congratulations on a very nice paper. I hope you found the review process to be constructive and are pleased with how the manuscript was handled editorially. We look forward to future exciting

submissions from your lab.

Sincerely,
